# Bioplastic Production from Agri-Food Waste through the Use of *Haloferax mediterranei*: A Comprehensive Initial Overview

**DOI:** 10.3390/microorganisms12061038

**Published:** 2024-05-21

**Authors:** Angela Longo, Francesca Fanelli, Marianna Villano, Marco Montemurro, Carlo Giuseppe Rizzello

**Affiliations:** 1Department of Environmental Biology, Sapienza University of Rome, 00185 Rome, Italy; angela.longo@uniroma1.it (A.L.); carlogiuseppe.rizzello@uniroma1.it (C.G.R.); 2Institute of Sciences of Food Production (CNR-ISPA), National Research Council of Italy, 70126 Bari, Italy; francesca.fanelli@ispa.cnr.it; 3Department of Chemistry, Sapienza University of Rome, 00185 Rome, Italy; marianna.villano@uniroma1.it; 4Research Center for Applied Sciences to the Safeguard of Environment and Cultural Heritage (CIABC), Sapienza University of Rome, 00185 Rome, Italy

**Keywords:** *Haloferax mediterranei*, bioplastics, polyhydroxyalkanoates, agri-food waste, bioreactor

## Abstract

The research on bioplastics (both biobased and biodegradable) is steadily growing and discovering environmentally friendly substitutes for conventional plastic. This review highlights the significance of bioplastics, analyzing, for the first time, the state of the art concerning the use of agri-food waste as an alternative substrate for biopolymer generation using *Haloferax mediterranei*. *H. mediterranei* is a highly researched strain able to produce polyhydroxybutyrate (PHB) since it can grow and produce bioplastic in high-salinity environments without requiring sterilization. Extensive research has been conducted on the genes and pathways responsible for PHB production using *H. mediterranei* to find out how fermentation parameters can be regulated to enhance cell growth and increase PHB accumulation. This review focuses on the current advancements in utilizing food waste as a substitute for costly substrates to reduce feedstock expenses. Specifically, it examines the production of biomass and the recovery of PHB from agri-food waste. Furthermore, it emphasizes the characterization of PHB and the significance of hydroxyvalerate (HV) abundance in the formation of Poly(3-hydroxybutyrate-co-3-hydroxyvalerate) (PHBV) copolymer. The downstream processing options are described, and the crucial factors associated with industrial scale-up are assessed, including substrates, bioreactors, process parameters, and bioplastic extraction and purification. Additionally, the economic implications of various options are discussed.

## 1. Introduction

Since its inception, plastic has contributed to innovation in several areas, including economic development and the progress of society [1]. Its resistance and its versatility, combined with its light weight, make it an appreciated choice among manufacturers and consumers, favored over other materials such as glass, wood, and metal [2].

Since 1950, the increase in plastic production has outpaced that of any other material, and its growth prospects are still persistent today with an expected growth rate of 3.2% from 2020 to 2027 [3]. Undoubtedly, the numerous advantages of plastic are accompanied by constant negative effects. The huge amount of plastic produced each year (more than 500 million tons worldwide) has a devastating impact on the environment [4]. Plastic leads to problems at every stage of its life, from production to years after its application has ended.

Some of the synthetic plastic materials of great economic/social importance are polypropylene (PP), polyethylene terephthalate (PET), polyvinyl chloride (PVC), and polylaminates (PI). None of these synthetic plastics are biodegradable. However, other synthetic ones are biodegradable, such as poly ε-caprolactone (PCL), polyurethanes (PUR), and polypropylene fumarate (PPF) [5,6].

The increased attention related to the environmental impact of plastic is leading to the limitation of single-use plastic products. Accordingly, the European Parliament implemented the European Union Directive EU 2019/904 by imposing limitations on the usage of various single-use plastic items, excluding those made from unaltered natural polymers. Growing concern about the sustainable use of natural resources and the need to reduce the environmental impact of plastic waste are pushing towards the rapid development of bioplastic materials based on renewable resources as a solution to global environmental challenges.

Bioplastics comprise a family of materials with different properties and applications. According to Lackner et al. [7], a plastic material is defined as a “bioplastic” if it is biobased, biodegradable in specific conditions, or has both properties. The bioplastic market is constantly growing. In 2010, 0.7 million tons of bioplastics were produced, versus 1.7 million tons in 2015 [8], and it has been estimated that annual bioplastic production will reach 3 million tons in 2025 [9]. Bioplastics’ advantages are multiple, including a reduction in fossil resource use, CO_2_ impact, and non-degradable plastic waste.

The food packaging industry is exploring a growing variety of biopolymers (e.g., polylactic acid, PLA, and bio-polyethylene, Bio-PE) which are classified as follows:Bioplastics extracted from biomass, which include polysaccharides naturally present for example in plants, including cellulose and starch, that are also treated and mixed with adjuvants to adapt to different types of uses [10];Bioplastics from bioderived monomers, like PLA, produced from lactic acid obtained chemically or via biomass fermentation, polyglycolic acid (PGA), obtained via the condensation or polymerization of glycolic acid and also produced via biomass fermentation, and bio-polyethylene, produced by starting with ethylene via biomass fermentation [10];Biocompounds, produced via the combination of two or more types of biopolymers to improve their physical and/or chemical characteristics [10];Bioplastics obtained and polymerized via microorganisms (e.g., *Azotobacter beijernickii, Cupriavidus necator*, and strains belonging to the *Bacillus*, *Pseudomonas*, *Rhodococcus*, *Micrococcus*, and *Rhododococcus* genera), which synthesize bioplastic naturally in response to excess carbon presence and nutrient-limited conditions.

Microbial fermentation is one of the most innovative biotechnologies to produce bioplastic. However, the overall high cost of such production represents an important obstacle to the industrialization of the production process and product commercialization. Bioplastic can be classified according to the substrates used as carbon sources (e.g., starch, cellulose, protein, lignin, or chitin) in the biotechnological process and in particular, starch-based biopolymers represent 50% of the global production of bioplastics. Among these, polybutylene succinate (PBS) and polycaprolactone (PCL, produced from fossil-based feedstocks but biodegradable) currently dominate the market [11]. Moreover, polyhydroxyalkanoates (PHAs) can be produced from different carbon sources (including waste organic streams) and stored in microbial cells in the form of granules within the cell cytoplasm as energy and carbon reserves to allow microorganisms to survive under environmental stress conditions. Such a transformation was recognized as a very proficient survival strategy used by several microorganisms [12].

In the near future, there will be an inescapable need to increase the food supply due to the rapidly growing world population. As food increases, a greater quantity of by-products will be produced. However, food by-products notably harbor substantial reserves of nutrients and bioactive compounds, thus allowing their utilization for the creation of novel food ingredients or products for human consumption that align with the principles of the circular bioeconomy [13].

Recent research has examined biotechnological applications, such as the identification of optimal fermentation conditions for microorganisms to produce PHA [4,9,12]. Currently, there is significant interest in reviewing the application of low-cost substrates as a source of nutrients for microbial growth and precursors to bioplastic production. The present review focuses on a particular microorganism, namely *Haloferax mediterranei*, which exhibits numerous advantageous characteristics and potential. In detail, this article covers the lack of a specific review concerning the production of bioplastics via *H. mediterranei* from food waste. In addition, a thorough comprehension of the pathways involved in the production of bioplastics and the critical points associated with industrial scale-up are assessed in order to highlight the current state of knowledge in this area and encourage further academic and industrial research.

## 2. Agri-Food Waste as a Substrate for Polyhydroxyalkanoate Production Using Pure Microbial Cultures

The utilization of food by-products will allow for a reduction in bioplastic production costs. Indeed, it was already reported that the final cost of PHA production can mainly be reduced through the proper selection of microorganisms and carbon substrates [14,15]. Food waste is a promising and inexpensive source of nutrients that contain the carbon, nitrogen, and phosphorus needed for the growth of microorganisms and the generation of PHA [16]. The use of agri-food waste for bioplastic production represents an efficient solution for reducing organic waste management’s impact and greenhouse gas emissions, thus introducing a low-cost and sustainable technology. However, these residues are usually formed via complex compounds and must be pretreated to allow their use as a substrate. The pre-treatment of agricultural waste provides a good quantity of simple sugars and fats usable for microorganisms [17]. For example, potato waste must be enzymatically hydrolyzed before being used as a substrate via *Bacillus circulans* for PHA production [18]. On the other hand, vegetable waste deriving from lemons, tomatoes, carrots, and fennel processing can be used as substrates for microbial growth without prior treatment. These products are favorable for the growth of extremophiles, such as thermophiles (*Bacillus thermantarcticus*) and alkalophiles (*Haloterrigena hispanica*), and the production of polyhydroxybutyrate (PHB) homopolymer [19]. Recently, wheat starch wastewater was used as an economic carbohydrate source for PHA production using a wild strain isolate of *Bacillus cereus* [20]. Examples of agri-food wastes used for different types of PHA-producer microorganisms are reported in Figure 1.

Among extremophiles, halophilic bacteria are promising microorganisms for the valorization of food waste. Halophiles can survive in hypersaline environments and were recently investigated as potential PHA-producing cell factories. *Haloferax mediterranei*, an archaeon that thrives in highly saline conditions, can metabolize different types of waste feedstocks. The growth in hypersaline substrates makes halophiles the most promising candidates for industrial applications [16]. The main advantages of using halophilic bacteria are, indeed, related to their need for high salt concentrations. Hypersaline substrates avoid contaminations, preventing the growth of non-halophilic microorganisms and, thus, reducing the costs of the energy input for substrates and bioreactors’ sterilization [21]. Moreover, due to the high intracellular osmotic pressure, these cells can be lysed in water, reducing PHA recovery costs [22]. Different salinities were tested in molasses wastewater to optimize *H. mediterranei*’s PHA productivity. The results showed that the maximum intracellular content of PHA was found at a salt concentration of 300 g/L, while the best microbial growth was obtained at 200 g/L, finally resulting in the best volumetric productivity in the latter [23]. In addition, external magnetic fields of 50 mT were tested at different salinity levels, demonstrating higher PHA volumetric productivity at 300 g/L with a comparable maximum growth rate to salinity at 200 g/L. The concentration of potassium and betaine in different conditions was also investigated, confirming their role in the regulation of *H. mediterranei*’s osmotic pressure [23].

*H. mediterranei* is probably the most preferred PHA producer among all the haloarchaea strains due to its high growth rate, metabolic versatility, and genetic stability, thus increasing the attention to new biotechnological applications for bioplastic production.

## 3. Genes and Pathways Involved in Bioplastic Production Using *H*. *mediterranei*

With the publication of the complete genome sequence of *H. mediterranei* in 2012 [24], the PHA biosynthetic gene cluster organization was described, and the key genes involved in poly(hydroxybutyrate-co-hydroxyvalerate) (PHBV) metabolism and regulation were identified and characterized [25,26,27,28,29,30,31]. Figure 2 summarizes the main genes and pathways involved in PHA production via *H. mediterranei*.

In *H. mediterranei*, six PHA granule-associated proteins have been identified; except for PhaZ1, which will be further described later, five of them are coded via the PHA biosynthetic cluster, which displays the conserved organization as *phaJ1*-*phaR-phaP-phaE-phaC* observed in other haloarchaea, indicating a functional relation between these five genes. The key enzyme of the cluster, which catalyzes the polymerization of 3-hydroxyacyl-coenzyme A (CoA) into PHAs, is the PHA synthase. In bacteria, PHA synthases are generally grouped into four classes according to their substrate specificities and subunit compositions [32]. The PHA synthase of *H. mediterranei* is homologous to the bacterial type III PHA synthase and is composed of the PhaE and PhaC subunits [31]. This species possesses four *pha*C genes, *pha*C, *pha*C1, *pha*C2, and *pha*C3. In the wild-type strain, with the exception of *phaC*, the other three genes are cryptic and not transcribed under PHA-accumulating conditions [33]. Moreover, *pha*C and *pha*E were mapped on the megaplasmid pHM300, and *pha*C1 was found to have a chromosome localization, whereas *pha*C2 and *pha*C3 were both located on the megaplasmid pHM500 [33].

The four PhaC proteins of *H. mediterranei* share an identity ranging from 43% to 58%, and they all contain catalytic triad residues (Cys-Asp-His). Both the “Lipase box-like” sequence (Gly-X-Cys-X-Gly-Gly) and the conserved motif of type III PHA synthase are present in PhaC, PhaC1, and PhaC3, while the C terminal sequence is missing in PhaC2, which has no PHA polymerization activity [33].

Chen and his coworkers [34] recently demonstrated that the deletion of a phosphoenolpyruvate synthetase (*pps)*-like gene activated the cryptic *phaC* genes. The PPS-like protein, which exhibited high homology with PPS enzymes, though it has evolved some distinct sequence features and functions [34], might act as a negative regulator of transcription. The authors also identified the key enzymes, one potassium-dependent pyruvate kinase (PYK_Hm_ [locus tag: HFX_0773]) and one phosphoenolpyruvate synthetase (PPS_Hm_ [locus tag: HFX_0782]), that catalyze the anabolic and catabolic directions of the PEP/pyruvate interconversion [34].

The *pha*H gene codes for the phasins, which are the major proteins present in the PHA granules. PhaP_HME_, despite sharing no homology with the bacterial PhaPs, which have been hypothesized to be implicated in the regulation of PHA synthesis, degradation, granule size control, and distribution [35], has a high alpha helix content, which resembles that of phasins in bacteria [25]. Also, in *H. mediterranei,* PhaP was demonstrated to be the major structural protein in PHA granules and to promote PHA accumulation and granule segregation [25].

The *phaR* gene codes for a PHA granule-associated regulator which directly binds to the promoter of *phaRP* and negatively regulate the transcription of this operon [26]. Different from what occurs in bacteria, the *pha*P gene overlaps by 8 bp that of the *pha*R gene, and these two genes are co-transcribed [25]. PhaR negatively regulates its own gene expression and the expression of *pha*P in the same operon [26]. In addition, in a PhaP-independent manner, PhaR can promote PHA synthesis and granule formation [26]. Downstream *phaR*, the *phaJ1 gene* codes for the only (R)-specific enoyl-CoA hydratases of the five identified in the genome of *H. mediterranei*, which is associated with PHA granules [29] (PhaJ1; locus tag: HFX_5217). The *phaJ2* to *phaJ5* genes are instead dispersed in the genome of *H. mediterranei*. PhaJ1 acts as the major enzyme in mediating PHA mobilization in *H. mediterranei* [30], linking PHA degradation to the β-oxidation pathway by catalyzing the dehydration of (R)-3HA-CoA to enoyl-CoA.

The sixth PHA granule-associated protein is PhaZh1, encoded by the gene *phaZh1* [29]. This gene is located near the *bdh*A gene (encoding putative 3HB dehydrogenase), forming a gene cluster (HFX_6463 to _6464) in *H. mediterranei*. PhaZh1 is an intracellular (lacking a signal peptide) PHA depolymerase with a patatin-like domain, while BdhA hydrolyzes t3HB monomers generated via PhaZ1 from natural PHA granules [29]. The deletion of the *phaZ1* gene has no significant effect on PHA catabolism in vivo, although in vitro, it has been recognized as a key enzyme of PHA hydrolysis, indicating that additional PHA polymerases might act during in vivo PHA degradation.

Outside the *pha* cluster, other primary enzymes are involved in PHA biosynthesis and metabolism:

(i) Two β-ketothiolases, PhaA and BktB, are responsible in *H. mediterranei* for delivering the precursors to PHBV biosynthesis [28]. PhaA has two subunits, α and β, one with the catalytic domain and one with the oligo-saccharide binding domain, coded respectively according to the co-transcribed genes *phaAα* and *phaAβ* (HFX_1023 and HFX_1022). PhaA catalyzes the condensation reaction (and the cleavage, as a reverse reaction) of two acetyl-coA units to form acetoacetyl-CoA or one acetyl- CoA and one propionyl-CoA to 3-keto valeryl-CoA, an initial step in PHB formation [28]. Similarly, BktB is encoded by two co-transcribed genes, *btkBα* and *btkBβ* (HFX_6004 and HFX_6003), and it consists of two subunits, the catalytic α and the oligo-saccharide substrate-binding beta. The catalytic triad “Ser-His-His” of the alpha subunits of Btkα and PhaAα, which is distinct from the bacterial “Cys-His-Cys”, is responsible for the substrate specificities.

(ii) Concerning *H. mediterranei*, Feng et al. [36] reported the presence in the genome of *H. mediterranei* of two genes coding for two putative homologues (PhaB1/B2, responsible for the 3HB-CoA and 3HV-CoA formation) of PhaB, the NADPH-dependent Acetoacetyl-CoA reductase that catalyzes the chiral reduction in acetoacetyl-CoA to (S) or (R)-3-hydroxybutyryl-CoA and was characterized in *H. hispanica* [37].

Up to now, *H. mediterranei* has been reported to synthesize poly(3-hydroxybutyrate-*co*-3-hydroxyvalerate) (PHBV) and poly(3-hydroxybutyrate-*co*-3-hydroxyvalerate-*co*-4-hydroxybutyrate) (PHBV4HB) [38,39]. Unlike most organisms, *H. mediterranei* does not require any 3HV precursor for PHBV synthesis. The biosynthetic pathway consists of the condensation of two acetyl coenzyme A (acetyl-CoA) molecules into acetoacetyl-CoA via BktB/PhaA, a reduction to (R)-3-hydroxybutyryl-CoA via PhaB, and the polymerization of (R)-3-hydroxybutyryl-CoA monomers via PhaC.

PHA granules are storing structures for carbon and energy in the presence of excess carbon sources in archaea [40]. In *H. Mediterranei,* four routes have been described [27] that could be employed to maximize propionyl-CoA production and store reducing power via PHBV synthesis when exposed to excessive carbon sources. On the contrary, under carbon starvation conditions, PHA is mobilized via hydrolysis through PHA depolymerase [29]. The 3HB monomers generated via PhaZh1 are then converted into acetoacetate through BdhA. PHA degradation occurs through the beta-oxidation pathway, which involves PhaJ and catalyzes the dehydration of (R)-3HA-CoA to enoyl-CoA.

Genetic manipulation strategies have also been used to improve PHBV production: by knocking out the exopolysaccharide (EPS) gene cluster, PHBV production was increased by 20% [41]. This percentage can be increased, reaching 70.46%, by adding the knock-out of the *pps*-like gene [34]. Meanwhile, downregulating the citrate synthase genes (*cit*Z and *glt*A) and redirecting the metabolic flux from the central metabolic pathways to PHBV synthesis [42], using a CRISPR-based interference (CRISPRi) approach, leads to an increase in PHBV levels of 76.4%.

In relation to the environmental conditions regulating PHA levels, it was demonstrated via proteomic analysis [43] that the PHA cell content increased linearly, along with increasing salinity, by overexpressing beta-ketoacyl-ACP reductase and 3-hydroxyacyl-CoA dehydrogenase while decreasing EPS production. Three pathways modulate PHA and EPS production in *H. mediterranei* under extreme salinity conditions [43]. The expression of 3-oxoacyl-ACP reductase and 3-hydroxyacyl-CoA dehydrogenase was found to be increased at high salinity, thus enhancing PHA biosynthesis. In addition, the enzymes serine-pyruvate transaminase and serine-glyoxylate transaminase were upregulated under these conditions, thereby increasing the glucose-to-PHA conversion yield. On the contrary, the enzymes sulfate-adenylyl transferase and adenylyl-sulfate kinase were downregulated under high salinity, thereby reducing EPS formation [43]. Thus, altering salinity was proposed [44,45] as a tool to direct the carbon flux toward the predominant PHA or EPS biosynthesis of *H mediterranei*. It was, moreover, reported that high salinity (250 g/L NaCl) inhibited EPS excretion, thus promoting PHA biosynthesis [45].

Achieving a comprehensive view of the proteins involved in these metabolic pathways and their regulation, as well as under different growth conditions, will be crucial in finding the best growing conditions and developing genetic manipulation approaches that can be used to drive biotechnological applications and maximize bioplastic production via this organism.

## 4. Bioplastic Production using *H. mediterranei* from Agri-Food Waste

Sugar and fatty acids are usually used via *H. mediterranei* as a source of energy and precursors to PHA, while inorganic salts are necessary for optimal microbial growth and production. In the literature, the first article about PHA production using inexpensive substrates focused on the use of starch. Nevertheless, Lillo and Rodriguez-Valera [46] concluded that the significant quantity of salts needed in the growth solution posed a significant challenge in utilizing *H. mediterranei* for PHB synthesis, as the expense of the inorganic salts was equivalent to that of starch [46].

A unique feature of *H. mediterranei* is its ability to also accumulate the PHBV copolymer when cultured on glucose, starch, or hydrolyzed whey as sole carbon sources in the absence of HV precursors (i.e., propionic and valeric acids) [21,47]. In general, the incorporation of HV units in the polymer chain improves its flexibility, resulting in lower crystallinity and higher toughness and elongation to breaking, with a consequent expanded processing range that is related to its HV content [48]. This is particularly relevant since the PHA properties allow for the biopolymer’s application in a wide range of industrial sectors, including packaging, agriculture, animal feed, disposable daily necessities, medicine, 3D printing, as well as smart material usages [49]. Also, it has been reported that PHA’s monomeric composition and crystallinity affect the biopolymer biodegradability rate, with the copolymers’ degradation in soil being faster than the homopolymers’ degradation due to lower crystallinity and a porous surface that facilitates the adsorption of microorganisms at the surface [50]. As an example, it was reported that, after 117 h under fed-batch fermentation in the presence of glucose and yeast extract (as carbon and nitrogen sources, respectively), *H. mediterranei* produced 85.8 g/L of dry cells with an intracellular PHBV content of 48.6% (*w*/*w*), holding between 10.7 and 12.3 mol% of the 3-HV unit in the chain structure [39]. A higher HV content, up to 43 mol%, was obtained by feeding *H. mediterranei* with a synthetic mixture of butanoic and pentanoic acids (accounting for 56 and 44 mol%, respectively), thus demonstrating the possibility of producing bespoke PHBV copolymers [51]. Examples of PHBV copolymer production with *H. mediterranei* using waste feedstocks are herein reported.

### 4.1. Cereal-Based Waste and Impact of Nutrient Limitation

*H. mediterranei* needs phosphorous to grow, but it can accumulate PHA under phosphorus-limiting conditions, and it is possible that phosphorus deficiency may affect the length of the HV chains in the copolymer structure [52]. As for the nitrogen source, a higher HV molar fraction was observed in the presence of ammonium than with nitrate in the culture medium, and lower C/N ratios allowed for producing richer HV polymers [53]. Chen et al. [54] demonstrated that the correlation between nitrogen limitation and PHA accumulation that is usually true for PHA-producing microorganisms is controversial for *H. mediterranei,* confirming the results previously reported by Lillo and Rodriguez-Valera [46]. The authors obtained a cell dry weight (CDW) and PHA production of 39.4 and 20 g/L (50.8%), respectively, using extruded corn starch and yeast extract (50 and 85 g/L, respectively) in fed-batch fermentation under a controlled pH after 3 days. Extruded corn flour was tested with extruded rice bran (50 g/L in total with a ratio of 1:8) and yeast extract (85 g/L) by Huang et al. [55] in a 5-L jar fermenter with a controlled pH and without nitrogen-limitation restrictions, obtaining 140 g/L of CDW and a yield (calculated as the percentage of PHA extracted from the recovered cell dry matter) of 55.6% after 5 days. Wasted bread (15% w/v) in water was used to prepare an extract and then supplemented with microfiltered sea water and 160 g/L of NaCl [22]. The pH was adjusted to 7.2 with 1 M ammonia solution. Different percentages of wasted bread extract and seawater were tested to select the best microbial growth evaluated spectrophotometrically, and the ratio of 40:60 was selected. After fermentation for 72 h at 37 °C in a 3L bioreactor with controlled stirring and a controlled pH, the PHBV obtained using different extraction methods was 21.6 ± 3.6 mg (a standard extraction/purification procedure with CHCl_3_:H_2_O mixture), 24.8 ± 3.0 mg (water-based extraction), and 19.8 ± 3.3 mg PHAs/g of wasted bread (water-based extraction followed by ethanol purification) [22]. In this case, the characterization of the obtained biopolymer, which presented an HV content ranging between 10.78 and 12.95% (wt/wt), showed lower melting-point values (between ca. 130 °C and 145 °C) than that (ca. 175 °C) of the pure PHB homopolymer, which was close to the decomposition temperature [22].

### 4.2. Wastewaters

Waste produced via dairy industries has been highly exploited for its valorization as a carbon source in bioplastic production, given that it represents one of the largest sources of wastewater in the agri-food sector. Koller [38,56] used whey, after the enzymatic hydrolysis of lactose into glucose and galactose, as a substrate in a 300-L fermenter, finding up to 13.63 g/L of biomass and 87.0% of PHA. Interestingly, the salty cell debris was collected and reused in a second fermentation step to avoid the high consumption of salt, but the CDW and PHA content was significantly reduced. Cheese whey was used also by Raho et al. [57] and Pais et al. [58]. In the latter study, by using hydrolysate cheese whey (obtained through a simple acid hydrolysis method), a volumetric productivity of 4.04 g PHBV/L/day, with intracellular polymer content of 53% (wt/wt), consisting of 1.5 mol% of HV, was obtained.

Another important wastewater is olive mill wastewater (OMW), which is mainly characterized by high amounts of polyphenols with high antibacterial properties. *H. mediterranei* exhibited significant cellular proliferation and showed resilience to the suppressive impact of polyphenols in a medium including different percentages of OMW, confirming the ability of some halophiles to grow in media containing phenolic compounds [59]. OMW was utilized as a feedstock for PHA production without the pretreatments (i.e., dephenolization and fermentation) that are typically employed when using mixed microbial cultures [60], and no inhibitory effects on the growth of *H. mediterranei* were observed at up to 25% OMW concentration, even though the optimal cell growth was observed at 5% OMW [61]. The optimized process was performed in a medium containing 15% OMW. The optimal conditions for achieving the maximum polymer yield (0.2 g/L) and PHA content (yield of 43%) were 37 °C, a stirring speed of 170 rpm, and 22 g/L of salt concentration [61]. The HV content in the stored polymer accounted for 6.5 mol%, a value very similar to that obtained (6 mol% of HV) with whey sugars as the sole carbon source, whereby a much higher intracellular polymer content (72.8%, PHA/cell dry mass) was achieved [38]. Notably, the copolymer produced with both feedstocks featured comparable thermal behavior, showing lower melting points (at 140.1 °C and 154.4 °C with OMW, or between 150 and 160 °C with whey sugars) than the melting point of the pure PHB homopolymer (ca. 175 °C), as previously reported.

Wastewater investigated for PHA production also includes sesame wastewater (SWW), which is produced during the process of removing the outer shell of sesame seeds. SWW was previously hydrolyzed using hydrochloric acid and then added to 100 g/L NaCl and 6.0 g/L yeast extract, reaching a final production of 0.53 gPHA/L. However, the addition of glucose generated higher PHA production (up to 20.9 g/L) compared to the system that was only supplied with SWW [62].

### 4.3. Algae, Vegetables, and Mixed Waste

Defatted biomass from the microalgae *Chlorella,* appropriately pre-treated with hydrochloric acid (HCl) to promote carbohydrate hydrolysis, was also used as a substrate for *H. mediterranei* growth, resulting in PHB production of 3.79 ± 0.03 g/L (55.5% of CDW) after 120 h of cultivation [63]. Ghosh et al. [64] increased the DCW and PHB amount produced using the green macroalga *Ulva* sp. at 3.8 ± 0.2 g/L and 2.2 ± 0.12 g/L, respectively, when *H. mediterranei* was grown in 25% (*w*/*w*) of *Ulva* sp. hydrolysate at 42 °C.

Date waste biomass extract was obtained from immature falling, rotten, spoiled, and low-grade fruit and used for bioplastic production via *H. mediterranei,* and an optimal concentration of key trace elements without any supplementation was found. CDW of 12.8 g/L and PHA content of 3.20 g/L were the highest production obtained at the laboratory scale, while a further scale-up in a 5 L bioreactor yielded CDW of 18.0 g/L and PHA of 25% [65]. Vinasse, the by-product obtained after the production of ethanol from sugarcane molasses, was used after pretreatment with 5.0 g of activated carbon at a pH of 2 and a concentration of 25 and 50% in MST medium (1416, DSMZ) for the production of PHBV via *H. mediterranei* DSM 1411 [66]. The results achieved a final production of 19.7 g/L of PHA with a high PHA/CDW ratio (70%) and volumetric productivity of 0.21 g/L/h. A higher production of PHA was found when 25% of pretreated vinasse was used, but the subsequent biopolymer characterization showed the best polymer composition in a 50% vinasse medium [66].

The use of a combined food waste substrate, consisting of 28% cooked rice, 32% cooked ground beef, and 40% chopped raw cabbage, was processed into a slurry in anaerobic conditions for three weeks, and the resulting liquid was used as the sole carbon source for PHBV production. The findings indicated that a combination of carboxylates produced during the waste fermentation resulted in a PHBV yield that was 55% more than that achieved with glucose [17]. In detail, the fermentation of the waste slurry obtained from food waste resulted in CDW and PHBV of 2.99 ± 0.29 g/L and 1.57 ± 0.05 g/L, respectively.

The main parameters (i.e., CDW and PHA intracellular content) obtained in relevant studies investigating biopolymer production via *H. mediterranei* with waste agri-food substrates are reported in Table 1.

Research on bioplastic synthesis via microorganisms, starting from substrates consisting of by-products, surplus, and waste from agri-food chains, represents an opportunity for development and technological innovation that is able to promote the sustainable management of the environment and provide greater attention to nature conservation.

## 5. Downstream Processing Options

Taken as a whole, the possibility to accumulate the PHBV copolymer without medium supplementation with HV precursors provides another important advantage for the potential use of *H. mediterranei* at the industrial large-scale level, together with the negligible risk for microbial contamination due to the high salt concentration and easy recovery of the polymer via osmotic shock, as mentioned above. In particular, the downstream process typically consists of several steps categorized into two main strategies, consisting of the use of appropriate organic solvents (e.g., halogenated compounds, alkanes, alcohols, esters, and ketones) to extract PHA granules from inside bacterial cells or additives/chemical and biological agents to disrupt the microbial cell wall and release the intracellular polymer. Both approaches are often coupled with pretreatments to enhance the permeability of the cellular membranes, followed by final polymer purification steps [67]. In particular, the downstream process can account for up to 50% of the total PHA production cost, mainly due to the use of large amounts of solvents and high energy requirements [68]. Therefore, several solvent-free extractions were tested with the aim of decreasing the environmental impact, the risk of the process related to the use of chemicals, and the production cost [58]. Interestingly, slightly higher hydroxyvalerate content (12.95 vs. 10.78%, *w*/*w*) was found in PHBV obtained through water-based extraction compared to the conventional approach [22]. Therefore, from a technological and applicative point of view, the opportunity to employ *H. mediterranei* while avoiding sterilization practices and simplifying the PHA extraction procedure (based on the salinity decrease in the external medium), along with the possibility to exploit low- or no-cost carbon sources as feedstock, would contribute to significantly reducing overall production costs.

## 6. Critical Points Related to the Industrial Scale-Up of PHA Production via *H. mediterranei*

The use of economical and widely available substrates, optimization for efficient fermentation, and downstream processing conditions, such as the correct choice of durable equipment and feeding strategies, can contribute to the full exploitation of *H. mediterranei*’s potential in industrial bioplastic production. Techno-economic analysis, facilitated via various informatics tools, is essential in estimating economic performance and guiding the development of large-scale sustainable processes.

### 6.1. Choice of a Substrate

As described above, the use of different substrates can significantly change the fermentation performance. It has been largely reported that the fermentation conditions promoting microbial cell growth, which is fundamental in obtaining good productivity, are often different from those necessary to stimulate PHA synthesis and accumulation. For this reason, the employment of substrates with different intrinsic characteristics presupposes a new optimization of the fermentation parameters. This is challenging, given that the use of feedstocks derived from agricultural or agro-industrial wastes/by-products instead of synthetic media is currently considered the most promising option to reduce the production costs of PHA [69,70]. The natural variability found in these matrices, moreover, represents an additional critical issue for the standardization of the downstream processes aiming at recovering and purifying PHAs. Indeed, it has been reported that changes in the C:N ratios of the substrates led to variations in polymer accumulation, thus requiring a continuous adaptation of the purification protocol [9]. In particular, the molar mass of the polymer strongly affects its technological properties, thus making the control of the biosynthesis fundamental from the early fermentation stage. A complete characterization of the chemical composition of the feedstocks, also including the evaluation of the processing conditions and the factors affecting its variability, represents the key factor from the perspective of PHA production standardization. Based on this deep characterization, it is possible to define the type and intensity of the pretreatments necessary for the conversion of the feedstocks into suitable fermentation media [69,70]. Moreover, the detoxification or dilution of certain agri-food wastes may be necessary to avoid the inhibition of microbial growth or PHA synthesis. Although the pretreatment of feedstock can be considered an additional cost, the overall cost of the production process is often lower than that related to the use of carbon sources selected ad hoc among pure chemical reagent. Additionally, the use of complex feedstocks represents a strategy for the valorization of wastes and by-products into value-added products [69,70].

### 6.2. Bioreactors

Bioreactors designed for the cultivation of PHA-producing microorganisms must be equipped with stirring systems installed at the top of the apparatus because they are typically sealed off from the external environment via a double mechanical seal that includes a rotating and static ring. The risk that the PHA-rich cells could infiltrate and form a PHA film between the two rings has been reported. As a consequence, a separation of the rings can occur, thus leading to losses of the liquid substrate in bioreactors stirred from below. Moreover, the use of a highly saline concentration often causes the formation of salt crystals between the two rings, thus leading to leakage through the bioreactor seals [71]. Due to the high salt concentration necessary in growing haloarchaea like *H. mediterranei*, highly corrosion-resistant materials must be used in bioreactor building. Stainless steel is largely used as a bioreactor material. Nevertheless, it is susceptible to corrosion via Cl^-^ ions in long-term use, which represents one of the main obstacles that limit its application in extreme halophile cultivation systems [72]. Several studies have proposed innovative, corrosion-resistant polymeric materials for bioreactor building. The plug flow-type bioreactor employed by Bhattacharyya et al. [66] for *H. mediterranei* cultivation comprised the polymer poly(methyl methacrylate) (PMMA). A corrosion-resistant bioreactor for extreme halophile cultivation was designed by Hezayen et al. [73] and made by the company FairMen Tec. using tech glass, poly(ether ether ketone) (PEEK), and silicon nitride ceramics. Aiming at *cultivating H. mediterranei,* Lorantfy et al. [74,75] manufactured an advanced version of this bioreactor by adapting a corrosion-resistant 2L-bioreactor made of borosilicate/PEEK (Labfors bioreactor; Infors, Switzerland). In particular, it included a borosilicate glass vessel, while corrosion-free glass and PEEK were used for all parts and devices in contact with the saline cultivation substrate. Moreover, all the connections were made with silicone, glass, or PEEK tubing [74]. The development of a cylindrical bubble-column bioreactor (Möstl Anlagenbau, Austria) was carried out by Mahler et al. [76]. The apparatus, having a working volume of 15 L, was designed for the continuous cultivation of extreme halophiles. The highly corrosion-resistant nickel–molybdenum alloy “Hastelloy” was used in building this bioreactor that was, moreover, equipped with an external cell retention unit, including a tangential flow filter (in polysulfone). The filter module allowed for the performance of the process at different dilution rates, thus increasing the polymer yield. Mahler et al. [76] reported that the bubble-column design was three times more efficient than a common, continuously stirred tank reactor since it required lower energy input to provide the desired oxygen (O_2_) level in the liquid substrate.

### 6.3. Feeding Strategies and Salinity Control

Several feeding schemes have been evaluated to assess the optimal utilization of the nutrients in the medium of PHA-producing microorganisms [44]. Among the different fermentation strategies, batch fermentation is the most common, thanks to its low operating costs and flexibility. However, the consumption of PHAs is possible in this condition, especially when the carbon source becomes limiting [77]. Moreover, while many organisms accumulate PHAs under non-growing conditions (e.g., *P. putida*) [78], organisms that produce PHAs during the growth phase, like *H. mediterranei* [53], are suitable for such types of continuous fermentation schemes.

Compared to batch fermentation, fed-batch processes present several advantages, including the prevention of PHA degradation and the accurate control of the metabolic intake, thus allowing for the modification of the PHA structure by varying the carbon source over time. Moreover, fed-batch fermentation allows online modifications to the medium formulation and fermentation parameters in order to improve the accumulation of PHAs. Fermentation schemes combining batch and fed-batch stages were also evaluated. A starting batch stage, optimized to provide the optimal growth conditions for the microorganism, was suggested [77,79], aiming at obtaining the desired microbial biomass. Fed-batch fermentation was proposed as a second stage of the process since it was the suitable strategy to promote the accumulation of PHAs. Indeed, it is possible to maintain, under a limited concentration, one or more essential nutrients, while the carbon source is continuously added to the media [77,79]. As mentioned before, the salinity of the cultivation medium can be modulated not only to obtain higher PHA concentrations but also to reduce losses in the carbon source of EPS formation associated with *H. mediterranei* cultivation and without the use of genetically engineered strains.

### 6.4. Optimization of Downstream Processing

Due to the intracellular nature of biopolymers’ accumulation, the downstream processing of PHAs represents a relevant production cost that adds to that of the substrate. Overall, following biosynthesis, the microbial biomass is first retrieved from the medium. This is typically accomplished through centrifugation and/or filtration. Subsequently, the PHA granules are extracted and separated from the cells. In most instances, PHAs must then undergo additional purification to eliminate any residual biomass. If high-purity PHAs are required, the use of chlorinated solvents or acetone in the extraction and purification stages is required. Enzymatic treatment, chemical treatments with sodium hypochlorite or surfactants, and mechanical disruption are other downstream processing methods that are available [9]. It must be considered that the purification processes translate into additional demands for energy and chemical substances, which negatively affect the costs and sustainability of bioplastic production. Substantial improvements in the purification processes for PHAs have been recognized as the key to improving the market competitiveness of these polymers [9,44]. In downstream processing, haloarchaea present advantageous features related to their adaptability to extreme salinity. In 1986, Fernandez-Castillo et al. [40] focused on haloarchaeal PHA research and demonstrated that *H. mediterranei* cells easily lyse in the absence of salt (hypotonic media like distilled water) and release PHA granules into the aqueous phase [40]; thus, the authors hypothesized about the development of sustainable downstream procedures. PHA granules have a lower density than water and any cell debris; therefore, they can be recovered via centrifugation, which leads to the formation of a viscous layer on the surface that can easily be removed. Nevertheless, the PHA granules recovered in this way are still wrapped in a membrane. Thus, further purification of the biomass is required, especially if highly pure PHAs are needed (e.g., for biomedical applications).

An extraction procedure for the PHA from *H. mediterranei* cells cultivated on wasted bread supplemented with seawater, and based on repeated washing with demineralized water, was successfully applied instead of the conventional procedure with chloroform by Montemurro et al. [22].

In a PHA production process based on the fermentation of waste rice stillage using *H. mediterranei* [80], the cultivation broth was recovered after the production phase to investigate potential recycling processes. In particular, the cell mass was separated after settling for 24 h, and the exhausted substrate, containing 222 g/L of NaCl, was desalinated through the addition of hot decanoic acid, which caused the precipitation of the pure salts. The desalted stillage and decanoic acid were then separated into two phases, and the decanoic acid phase was used for subsequent desalination cycles. The recovered salts were reused for substrate supplementation. PHA recovered from the cell biomass was carried out in a lysis tank, under stirring, in a sodium dodecyl sulfate solution. The PHAs were then purified using an organic solvent and dissolved in hot chloroform. After cooling and polymer precipitation, the chloroform was recycled [80]. An efficient process for recovering high-purity PHB produced via *H. mediterranei* through the use of acetone as the sole extraction solvent was proposed [81]. Acetone can be produced from several biological biomasses via anaerobic fermentation with solventogenic *Clostridia*, and its use is, therefore, considered more sustainable than the largely used solvent chloroform [82]. However, since PHA is not soluble in acetone at room temperature and ambient pressure, these conditions are not suitable for the extraction of PHAs from *H. mediterranei* cells since it is accumulated in a crystalline structure. Nevertheless, under high-temperature and -pressure conditions (121 °C, 7 bar) in a confined environment, PHAs were easily dissolved from the *H. mediterranei* biomass. PHAs in a highly pure form were then separated through precipitation from the solution after cooling to room temperature. Cell debris was separated from the hot solution through a filtration unit under moderate pressure. After the PHAs’ precipitation, the co-extracted lipids remained in the acetone solution. The acetone was reused for other extraction cycles after distillation, aimed at lipids’ removal [81]. Lipids could be used for biodiesel synthesis, according to protocols already proposed for similar raw materials [83].

## 7. Environmental Systems and Techno-Economic Analysis for Industrial Applications

Environmental system analysis tools are commonly used to examine social, technical, and natural systems and the links between them, while techno-economic studies are applied to evaluate the industrial feasibility of processes through the identification and analysis of the key parameters affecting the production cost, thus identifying bottlenecks and guiding researchers towards the development of cost-effective processes. The combination of the two types of evaluations applied to the PHA production process is retained essentially to predict the sustainability of the process within a circular economy context. The most common analytical tool is represented in the Life Cycle Assessment [84]. However, attempts to find and quantify the environmental impact of PHA production via LCA tools have focused on isolated aspects of production, such as energy requirements or CO_2_ emissions [85]. Therefore, an analysis including the complete process and all the parameters is required [84]. Environmental Impact Assessment (EIA), Ecological Risk Assessment (ERA), and Material Flow Analysis (MFA) were also applied to PHA processes [84], but only a few studies have considered an *H. mediterrainei*-based process as the target of their investigation.

The cost evaluation needs to take into account several factors, which requires a case-by-case examination, considering the complexity of the process. Figure 3 summarizes the main factors that drive PHBV manufacturing costs.

The price of production for 1 kg of poly-3-(HB-co-HV) was calculated to be 2.82 EUR at a volumetric productivity of 0.29 g/L h when using hydrolyzed whey as a carbon source for *H. mediterranei*. This price is significantly lower than that calculated to produce PHA via recombinant *E. coli* at ~4.0 EUR [21]. A techno-economic analysis of the process based on *H. mediterranei* and the waste stillage of a rice-based ethanol industry [80], for example, estimated the PHA cost at 2.05 USD/kg for annual production of 1890 tons and identified desalination as the main cost-impacting factor. However, the general price for the production of PHB (between around 2 to 5 EUR/kg) is presently not competitive with the cost of standard petrochemical plastics (less than 1 EUR/kg) [68]. However, several issues need to be resolved in order to make the technology feasible, such as the difficulty of working with highly saline media and the generation of highly saline residues after polymer recovery [58]. Although not considered a conventional pollutant, the abundant release of high-salinity wastewater into the environment can negatively affect aquatic life and agricultural soil fertility [86]. The recovery of salt within the production cycle, or the correct management of brine disposal, must, therefore, be taken into consideration from the plant engineering point of view and in the evaluation of the production costs of PHB with halophiles.

## 8. Conclusions and Perspectives

Biotechnologies to produce PHAs and other biobased chemicals are rapidly improving. Thanks to the development of new additives, blends, and processing technologies, it is expected that a high amount of PHAs will be available on the global market in the future. Despite efforts to make PHA production more competitive, higher production costs in comparison to petroleum-derived plastics hamper its larger diffusion.

There is a pressing need to enhance both the sustainability and economic feasibility of industrial PHA production. This can be achieved by utilizing more cost-effective and efficient carbon sources, along with the development of purification processes that are both more environmentally friendly and efficient. In this context, *H. mediterranei*, which presents all the “green” advantages of PHAs synthesizing haloarchea, together with great adaptability to growth substrates, can be considered one of the most promising tools for a large scale-up of bioplastic production processes.

## Figures and Tables

**Figure 1 microorganisms-12-01038-f001:**
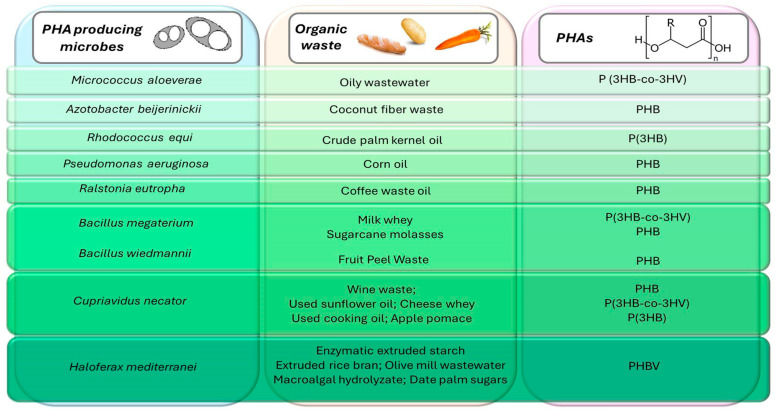
Schematic representation of the most important pure culture of microorganisms used in bioplastic production with organic waste as inexpensive substrates, including the type of polymer synthesized.

**Figure 2 microorganisms-12-01038-f002:**
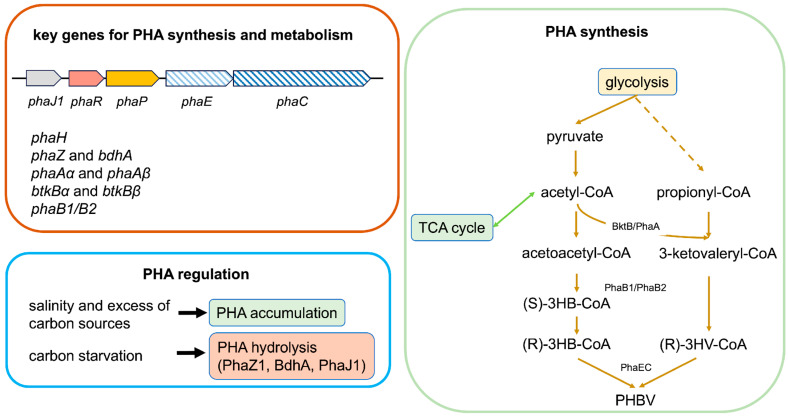
Schematic representation of key *H. mediterranei* genes involved in PHBV synthesis, metabolism, and regulation. *phaJ1:* enoyl-CoA hydratases J1; *phaR*: PHA granule-associated regulator; *phaP*: granule structural protein; *phaE*: subunit E of PHA synthase; *phaC*: subunit C of PHA synthase; *phaH*: phasin; *bdhA*: 3-hydroxybutyrate dehydrogenase *phaZ*: PHA polyhydroxyalkanoate depolymerase; *phaAα*: α subunit of β-ketothiolases PhaA; *phaAβ*: β subunit of β-ketothiolases PhaA; *btkBα*: α subunit of β-ketothiolases Btk B; *btkBβ*: β subunit of β-ketothiolases BtkB; *phaB1*: acetoacetyl-CoA reductase B1; *phaB2*: acetoacetyl-CoA reductase B2; see text for details.

**Figure 3 microorganisms-12-01038-f003:**
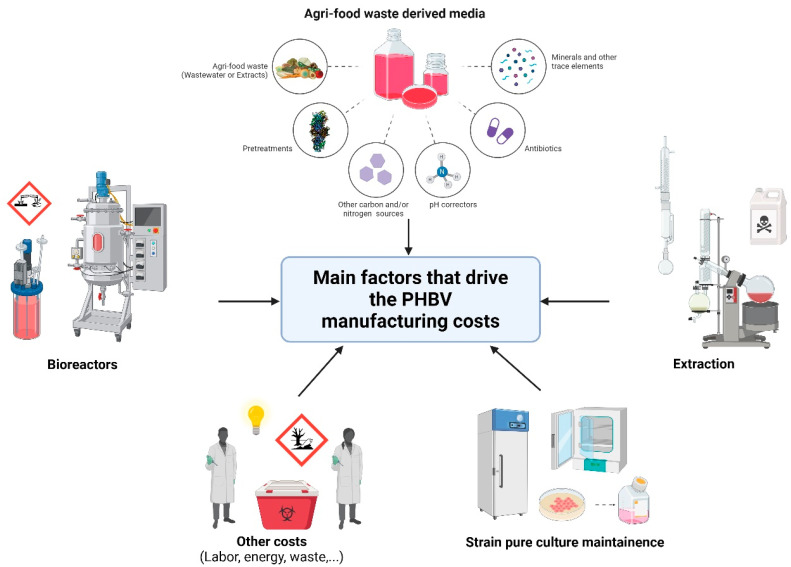
Graphical representation of the most important factors that drive PHBV production costs.

**Table 1 microorganisms-12-01038-t001:** Cell dry weight (CDW) and PHA percentage (calculated as the percentage of PHA extracted from recovered cell dry matter) produced via *H. mediterranei* in media prepared from different by-products, surplus, and waste from agri-food chains.

Carbon Source	CDW (g/L)	PHA (%)	References
Enzymatic extruded starch	39.4	50.8	[54]
Extruded rice bran	140.0	55.6	[55]
Hydrolyzed whey permeate	13.6	66.0	[56]
Cheese whey hydrolysate	7.5	53.0	[58]
Ricotta cheese whey	18.3	7.0	[57]
Olive mill wastewater	0.2	43.0	[61]
Macroalgal hydrolyzate	3.8	57.9	[64]
Defatted Chlorella biomass	3.8	55.5	[63]
Date palm sugars	18.0	25.0	[65]
Fermented food waste	2.99	52.5	[17]
Sesame seed wastewater + glucose	50.0	75.0	[62]
Vinasse	19.7	70.0	[66]
Wasted bread	6.7	24.0	[22]

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
