# Peer review of "Bioplastic Production from Agri-Food Waste through the Use of Haloferax mediterranei: A Comprehensive Initial Overview"

_microorganisms, 2024, doi:10.3390/microorganisms12061038_

Round 1
Reviewer 1 Report
Comments and Suggestions for Authors
The repetition rate of the manuscript has reached 40%, please ask the author to reduce the repetition rate of the manuscript.
Title: The main text of the manuscript is agri-food waste, not food waste. What is “first”? There are too many reviews about PHA production.
Abstract: Please highlight the innovation of this study. Please define abbreviations of “HV” and “PHBV”. Similarly, the author only talked about food waste.
The manuscript lacks the section of “Introduction”, please supplement. The Introduction should include the necessity of writing the manuscript, current research progress, logic, and originality. The author can refer to the literature. https://doi.org/10.1016/j.envres.2023.117796.
Line 75-78. Why is the “spp”?
Hfx. mediterranei should be “H. mediterranei”. Please have the author carefully review the manuscript.
Line 137-139. Why are only few research on the use of specific strains to produce PHA from fermented food waste?
Fig. 1 is too simple. How to reflect regulation? What is the full name of the key gene?
Line 185-257. The logic is quite chaotic. The author should provide more results and conclusions, rather than extensive xx et al., found, reported.....
Line 278: 56 and 44 mol%???
Similarly, The logic is quite chaotic in Section 4. Please rewrite.
Table 1: the content of PHA reached 87%. It looks different from the usual result.
Please rewrite the Section of “8. Conclusion and perspectives”. Especially the perspectives.
There are still some formatting and grammar errors in the manuscript. Please check the entire text carefully and correct the formatting errors.
Comments on the Quality of English LanguageThe manuscript is difficult to read. Extensive editing of English language required
Author Response
Dear reviewer,
thank you for your suggestion. We revised the manuscript according to your suggestion. Following a point-by-point response:
The repetition rate of the manuscript has reached 40%, please ask the author to reduce the repetition rate of the manuscript.
Thank you for the suggestion. We rewrote the section containing repetitions to reduce the repetition rate of the manuscript.
Title: The main text of the manuscript is agri-food waste, not food waste. What is “first”? There are too many reviews about PHA production.
Thank you for this comment. We modified the Manuscript title into “Bioplastic production from agri-food waste through the use of Haloferax mediterranei: a first overview”.
We would like to underline that we agree with the reviewers that there are many reviews about PHA production, but the main topic of our manuscript is the evaluation of the state of the art of the production of PHA focusing only at one microorganism (H. mediterranei) and the use of low-cost substrates derived from agri-food waste. In our opinion this is the difference between our manuscript and the other reviews already published. We explained this reason in the abstract and introduction section as suggested by reviewer.
Abstract: Please highlight the innovation of this study. Please define abbreviations of “HV” and “PHBV”. Similarly, the author only talked about food waste.
Thank you for the suggestion. We modified the abstract accordingly.
The manuscript lacks the section of “Introduction”, please supplement. The Introduction should include the necessity of writing the manuscript, current research progress, logic, and originality. The author can refer to the literature. https://doi.org/10.1016/j.envres.2023.117796.
Thank you. We modified the first section according to reviewer suggestion. The section title has been changed into “introduction”. The paragraph describing the current research progress about this topic and the originality and logic of this paper have been added.
Line 75-78. Why is the “spp”?
Thank you for the suggestion, we modified the sentence accordingly. The main species able to polymerize bioplastic have been reported also in Figure 1.
Hfx. mediterranei should be “H. mediterranei”. Please have the author carefully review the manuscript.
Thank you for the comment. The abbreviation for the genus was adjusted according to reviewer suggestion. We used the abbreviation Hfx in the first submission considering that hundreds of papers in literature utilize the abbreviation Hfx or H in alternative forms.
Line 137-139. Why are only few research on the use of specific strains to produce PHA from fermented food waste?
Thank you for the comment. The sentence has been modified “H. mediterranei is probably the most preferred PHA producer among all the haloarchaea strains due to its high growth rate, metabolic versatility, and genetic stability, thus increasing the attention in new biotechnological application for bioplastic production.”
Fig. 1 is too simple. How to reflect regulation? What is the full name of the key gene?
Dear reviewer, the Fig. 1 is a schematic representation of the H. mediterranei key genes involved in PHBV synthesis, metabolism and regulation. We believe that for a review this schematization is appropriate to give the readers an overview of the genes and regulation of PHA metabolism. We have included in the manuscript many references to deepen this topic, which was exhaustively addressed by other authors with specific attention to specific environmental condition that can regulate PHA metabolism. Gene names are those included in the figure. Eg. phaJ1 is the name of the gene encoding the enoyl-CoA hydratases J1. However, we included the product encoded by the corresponding gene in the caption.
Line 185-257. The logic is quite chaotic. The author should provide more results and conclusions, rather than extensive xx et al., found, reported.....
Thank you for the suggestion. We extensively modified the paragraph accordingly.
Line 278: 56 and 44 mol%???
Thank you for the comment. We checked the sentence and modified accordingly the text.
Similarly, The logic is quite chaotic in Section 4. Please rewrite.
Thank you for the suggestion. We modified the section in subsection describing the valorization of different types of agri-food waste in different paragraph. In addition, we rewrote the first paragraph of the section highlighting the parameters usually evaluated in research articles evaluating the production of PHBV from waste.
Table 1: the content of PHA reached 87%. It looks different from the usual result.
Thank you for the suggestion. We have checked the PHA values and corrected accordingly.
Please rewrite the Section of “8. Conclusion and perspectives”. Especially the perspectives.
Thank you for the suggestion. We rewrote the section accordingly.
There are still some formatting and grammar errors in the manuscript. Please check the entire text carefully and correct the formatting errors.
Thank you for the suggestion. We corrected and double checked all the formatting errors in the manuscript.

Reviewer 2 Report
Comments and Suggestions for Authors
The article "Bioplastic production from food waste through the use of Haloferax mediterranei: a first overview" is interesting and fits the topic of the journal, but requires a minor revision.
Comments:
1. Introduction page 2 verse 75-78: - it is worth adding a graphical chart: showing the types bioplastics obtained and polymerized by microorganisms (e.g. Azotobacter beijernickii, Bacillus spp., Pseudomonas spp., Rhodococcus spp., Ralstonia eutropha, Micrococcus spp., and Rhododococcus spp.) and closing the bracket.
2. 2.Microorganisms for innovative bioplastic production: add a table specifying the name of biowaste / microorganisms / type of polymer produced; it will be more clearer for readers
3. 7. Environmental systems and techno-economic analysis for industrial applications: This point requires clarification and supplementation with a graphical presentation of PHA production prices from specific renewable sources and comparison to commercial production.
Comments on the Quality of English LanguageMinor editing of English language required
Author Response
Dear reviewer,
thank you for your suggestion. We revised the manuscript according to your suggestion. Following a point-by-point response:
Introduction page 2 verse 75-78: - it is worth adding a graphical chart: showing the types bioplastics obtained and polymerized by microorganisms (e.g. Azotobacter beijernickii, Bacillus spp., Pseudomonas spp., Rhodococcus spp., Ralstonia eutropha, Micrococcus spp., and Rhododococcus spp.) and closing the bracket.
Thank you for the suggestion. Figure 1 has been added in the modified version of the manuscript.
Microorganisms for innovative bioplastic production: add a table specifying the name of biowaste / microorganisms / type of polymer produced; it will be more clearer for readers
Thank you for the suggestion. Considering that in this section we introduced the importance of the use of food waste and the fermentation by microorganisms focusing on H. mediterranei, which was then exploited in the following sections, we added the information about examples of biowaste used by microorganisms in the figure included in the previous section (Figure 1).
Environmental systems and techno-economic analysis for industrial applications: This point requires clarification and supplementation with a graphical presentation of PHA production prices from specific renewable sources and comparison to commercial production.
Thank you for the suggestion. We included Figure 3 describing the most important factors affecting the production costs of PHA. The specific evaluation of the cost related to the production of commercial PHA and the comparison of production processes using alternative renewable sources needs specific skills in economy and we think it could be out of the topic of this manuscript, requiring a case-by-case evaluation.

Round 2
Reviewer 1 Report
Comments and Suggestions for Authors
Accept in present form.
Comments on the Quality of English LanguageModerate editing of the English language required.